# Rho-Kinase Inhibition of Active Force and Passive Tension in Airway Smooth Muscle: A Strategy for Treating Airway Hyperresponsiveness in Asthma

**DOI:** 10.3390/biology13020115

**Published:** 2024-02-11

**Authors:** Yuto Yasuda, Lu Wang, Pasquale Chitano, Chun Y. Seow

**Affiliations:** 1Centre for Heart Lung Innovation, St. Paul’s Hospital, Providence Health Care, University of British Columbia, Vancouver, BC V6Z 1Y6, Canada; yuto.yasuda@hli.ubc.ca (Y.Y.); lu.wang@hli.ubc.ca (L.W.);; 2Department of Pathology and Laboratory Medicine, University of British Columbia, Vancouver, BC V6Z 1Y6, Canada

**Keywords:** rho-kinase, asthma, smooth muscle tone, cytoskeletal stiffness, mechano-pharmacology

## Abstract

**Simple Summary:**

The effects of a rho-kinase inhibitor alone and in combination with a bronchodilator in the presence of a small-amplitude oscillatory strain on the mechanical properties of airway smooth muscle were described in this brief review. Findings from relevant studies provide a strong basis for human trials using rho-kinase alone or in combination with other interventions in the treatment of asthma.

**Abstract:**

Rho-kinase inhibitors have been identified as a class of potential drugs for treating asthma because of their ability to reduce airway inflammation and active force in airway smooth muscle (ASM). Past research has revealed that, besides the effect on the ASM’s force generation, rho-kinase (ROCK) also regulates actin filament formation and filament network architecture and integrity, thus affecting ASM’s cytoskeletal stiffness. The present review is not a comprehensive examination of the roles played by ROCK in regulating ASM function but is specifically focused on passive tension, which is partially determined by the cytoskeletal stiffness of ASM. Understanding the molecular basis for maintaining active force and passive tension in ASM by ROCK will allow us to determine the suitability of ROCK inhibitors and its downstream enzymes as a class of drugs in treating airway hyperresponsiveness seen in asthma. Because clinical trials using ROCK inhibitors in the treatment of asthma have yet to be conducted, the present review focuses on the in vitro effects of ROCK inhibitors on ASM’s mechanical properties which include active force generation, relaxation, and passive stiffness. The review provides justification for future clinical trials in the treatment of asthma using ROCK inhibitors alone and in combination with other pharmacological and mechanical interventions.

## 1. Introduction

Asthma is a complex disease with multiple causes. The fact that a beta2 agonist can relieve acute asthma exacerbation indicates that the dysfunction of airway smooth muscle (ASM) is one of the causes [1]. It is known that ASM is the main effector of the exaggerated airway narrowing that is responsible for the bulk of acute asthma symptoms and that chronic airway inflammation could underlie ASM dysfunction [2]. However, airway hyperresponsiveness associated with asthma is only partially attenuated even when airway inflammation is treated successfully [3], suggesting that it is related to a fundamental structural or functional change in the muscle and/or airways. Although ASM is central to the pathogenesis of asthma and airway hyperresponsiveness, it is unclear how much, and which exact changes in the muscle, contribute to these phenotypes.

It has been observed long ago that in non-asthmatic subjects, experimentally induced bronchoconstriction can be reversed by allowing the subjects to take deep inspirations (DIs) [4,5]. A DI refers to a full lung inflation, i.e., from functional residual capacity (FRC) to total lung capacity (TLC). The bronchodilatory effect of DI is, however, largely diminished in asthmatic subjects [6], especially in subjects with severe asthma [7]. The distinct difference in the responses between asthmatic and non-asthmatic subjects to a deep inspiration (DI) has provided us with a unique window of opportunity to look into the mechanisms underlying asthma pathophysiology. DIs taken before bronchochallenge are also known to reduce the subsequently induced bronchoconstriction in non-asthmatic subjects; this is known as the bronchoprotective effect of DI, which is also diminished or abolished in asthmatic subjects [8,9,10]. Big breaths with amplitudes less than a full lung inflation have been shown to be ineffective in preventing bronchoconstriction in subsequently challenged subjects; to observe the bronchoprotective effect of DI in non-asthmatic subjects, a prior full-amplitude DI is required [11]. This suggests that there is a threshold in the amount of lung inflation only above which the bronchoprotective effect of DI can be observed. This raises an interesting speculation that this threshold is higher in asthmatics. This higher threshold could be a result of altered ASM mechanical properties, stiffer airways due to airway remodeling, or weakened airway-parenchymal coupling due to inflammation, among other causes [2].

Enhanced signaling in the rho-kinase (ROCK) pathway has been implicated in the pathophysiology of asthma. There are many excellent reviews on the topics of ROCK involvement in airway hyperresponsiveness, airway remodeling, and lung inflammation [12,13,14,15,16,17,18,19]. The present mini-review focuses on the evidence supporting the hypothesis that the lack of a DI-induced bronchoprotective effect in asthmatics is at least partially caused by a change in the ROCK-regulated passive stiffness of ASM, which stems from the structural properties of the cytoskeleton of ASM and is separated from the force-generating actomyosin contractile apparatus. A corollary of the hypothesis is that passive ASM stiffness can be a drug target for treating asthma [20].

## 2. Canonical Models of Signaling Pathways Regulating ASM Contraction and Cytoskeletal Stiffness

A giant step forward in our understanding of smooth muscle activation and contraction was the discovery that phosphorylation of the 20 kilo-Dalton regulatory myosin light chain (MLC20) is a key step in the activation of myosin crossbridges and the ensuing actomyosin interaction that produces muscle contraction [21,22]. Subsequently, a calcium-dependent activation pathway has been established [23]. That is, stimulation of smooth muscle leads to an increase in the intracellular calcium concentration; calcium binds to calmodulin and myosin light chain kinase (MLCK); the calcium–calmodulin–MLCK complex, which is the active form of the MLCK, then phosphorylates MLC20. Activation of smooth muscle through the G-protein-coupled receptors (GPCR) also activates ROCK, which inhibits myosin light chain phosphatase (MLCP) and indirectly enhances MLC20 phosphorylation (Figure 1).

Inhibition of MLCP is not the only function of ROCK. As depicted in Figure 1, ROCK is an up-stream enzyme that regulates many branches of signaling pathways not associated with MLC20 phosphorylation in smooth muscle. ROCK is activated by the small GTPase RhoA. One of the downstream kinases from ROCK is LIM-kinase (LIMK). Phosphorylation of LIMK by ROCK activates the kinase, which in turn phosphorylates cofilin thus inhibiting cofilin’s actin-filament-severing function [24]. As a result, ROCK, through LIMK, promotes actin filament polymerization and strengthens the actin cytoskeleton in smooth muscle.

Phosphorylation of p21-activated kinase (PAK) by ROCK is another pathway that regulates actin polymerization and cytoskeletal stability in smooth muscle (Figure 1). Paxillin phosphorylation by PAK leads to a sequence of events resulting in the activation of the small GTPase Cdc42, which in turn activates the neuronal Wiskott–Aldrich syndrome protein (N-WASp) and finally actin polymerization by the actin-related-protein-2/3 complex (Arp2/3) [25].

Figure 1 depicts an abbreviated version of the signaling pathways regulating smooth muscle activation. More comprehensive descriptions of the pathways can be found in many review articles such as the one by Puetz et al. [26].

## 3. Two Compartments of ASM

The active force generated by ASM has been shown to be directly proportional to the amount of MLC20 phosphorylation when other force-maintenance mechanisms are excluded [27]. One of the force-maintenance mechanisms is the latch–bridge mechanism [28]. More recent studies have revealed the critical role of the cytoskeleton in the maintenance of actively developed force in ASM [29,30,31,32,33,34]. Interestingly, these studies also showed that disruption of cytoskeletal integrity can lead to a reduction in the externally measured active force without affecting the degree of MLC20 phosphorylation. The results suggest that the contractile machinery in smooth muscle can be divided into two compartments. One of them is the contraction or force-generating mechanism consisting of myosin and actin filaments that slide relative to each other due to the cyclic interaction of myosin crossbridges with actin filaments. For this mechanism to function, the myosin crossbridges need to be activated, i.e., phosphorylated.

The other compartment is the cytoskeleton which structurally supports the actomyosin contractile units and also provides mechanical coupling for force transmission within and outside of a muscle cell [35]. Although both compartments are crucial for the functions of smooth muscle, such as constricting an airway or blood vessel, the contractile domain has historically received more attention from researchers than the cytoskeletal domain has. This is also reflected in the treatment of diseases related to smooth muscle dysfunction. For example, the use of beta2 agonists for asthma treatment and angiotensin-converting enzyme inhibitors and calcium channel blockers for treating hypertension. Drug targets in the cytoskeletal domain have largely been ignored.

## 4. “Passive” Tension in ASM

When smooth muscle tissue is stretched, tension rises even when the muscle is not activated. This tension is traditionally called passive tension and is thought to stem from the extracellular matrix within which the muscle cells are embedded [36]. This notion is challenged by the observation that in ASM tissue preparations there is a calcium-sensitive component of the passive tension [37]. That is, when a relaxed ASM tissue is incubated in a zero-calcium Krebs solution, the muscle becomes more compliant; and when calcium is reintroduced to the Krebs solution, muscle stiffness is restored, even though the muscle is relaxed and without resting tension. To determine whether this calcium-sensitive component of stiffness is intracellular or extracellular in origin, Raqeeb et al. [37] used a non-specific calcium channel blocker (SKF-96365) to block calcium entry into the muscle cells incubated in a calcium-containing Krebs solution; they found that the ASM tissue preparation became more compliant, demonstrating that the calcium-sensitive component of the muscle stiffness resides within the ASM cells. More interestingly, they showed that the calcium-sensitive “passive” stiffness can be inhibited by ROCK inhibition, and yet totally insensitive to inhibition of MLCK. This is consistent with the notion that active force and cytoskeletal stiffness are regulated by separate signaling pathways, such as that depicted in Figure 1, and suggests that the ROCK-sensitive “passive” stiffness is related to the integrity of the cytoskeleton.

Tissue preparations of ASM, such as a tracheal smooth muscle strip or a bronchial ring, include nerve endings containing acetylcholine (ACh). Calcium can trigger the release of ACh from the nerve endings, leading to activation of GPCR by ACh. To determine whether the calcium-sensitive stiffness of relaxed ASM is mediated by ACh, Lan et al. [38] used a membrane-permeated trachealis preparation which was severely “skinned”, resulting in extensive loss of myosin from the muscle cells. They showed that the actin filaments were largely retained inside the skinned ASM cells. Therefore, they essentially created a cytoskeletal preparation where the cytoskeletal stiffness can be measured without the “contamination” of active stiffness stemming from actomyosin interaction. Another useful feature of the preparation was that it still contained cell membranes (even though punctuated with holes) with functioning GPCR embedded. With such a preparation, it has been shown that cytoskeletal stiffness increases with ACh stimulation even in the absence of calcium, suggesting that the calcium-sensitive component of the passive stiffness in ASM tissue preparations can be mediated by ACh activation of GPCR. More interestingly, this zero-calcium, ACh-sensitive stiffness can be inhibited by ROCK inhibitors but is insensitive to MLCK inhibition [38]. If the zero-calcium, ACh-sensitive stiffness can be used as a surrogate measurement for cytoskeletal stiffness, the results suggest that ROCK is a key enzyme that regulates the cytoskeletal integrity and its ability to structurally support the contractile machinery and to transmit force.

## 5. Maintenance of Active Force in ASM Regulated by ROCK in the Mechanically Dynamic Environment

ASM resides in a living breathing lung and is constantly under cyclic stress and strain due to tidal breathing and occasional DIs. It has been shown that the amount of stretch experienced by ASM in DIs is substantial and it results in a temporary but significant loss of the ability of ASM to generate force [39]. Breathing is therefore an important mechanism in the prevention of maximal contraction of ASM and thus in the maintenance of airway patency in a healthy lung. In vitro experiments have shown that ROCK is critical in the maintenance of actively developed force in ASM. In the presence of a ROCK inhibitor, ASM is not able to maintain the actively developed force even with the continuous presence of a contractile stimulus, especially when the muscle is under cyclic strain mimicking tidal breathing [40]. Studies by Wang et al. [41,42] corroborate the findings, where they found synergistic effects of a ROCK inhibitor combined with small amplitude length oscillation (mimicking tidal breathing) that resulted in ASM relaxation. The findings suggest that ROCK plays an important role in strengthening ASM contractility and making the muscle less susceptible to mechanical perturbations that tend to reduce contractility. This role makes ROCK precarious in that overprotection of the muscle from the relaxant effect of cyclic strain may reduce the beneficial effects of DIs in bronchodilation and bronchoprotection.

## 6. Overactive ROCK Signaling in Asthma

It is well-known that RhoA/ROCK is involved in the pathophysiology of asthma, especially in airway hyperresponsiveness, airway remodeling and pulmonary inflammation. In an ovalbumin-sensitized rat model of asthma, it has been shown that the expressions of RhoA and ROCK mRNA and proteins are significantly increased [43]. Increased protein expression of RhoA, ROCK, MLC20, and MLCK has also been found in the lungs of a murine model of asthma [44,45]. In human asthmatics, the amount of ROCK in the peripheral blood has been found to be elevated [46]. More recently the protein expression of both isoforms of ROCK, ROCK1, and ROCK2, have been measured in airways and lung section from human asthmatics and age-matched controls. Both isoforms were found to be elevated in the bronchial smooth muscle and in pulmonary arterial smooth muscle from human asthmatic lungs [47]. It appears, therefore, that over-active ROCK signaling is a consistent feature of asthma pathology.

In a mouse (ovalbumin-sensitized) model of asthma, it has been shown that inhibition of ROCK resulted in a reduction in lung resistance and suppression of airway hyperresponsiveness [44,48,49]. In ovalbumin-sensitized guinea pigs, ROCK inhibitor reversed histamine- and PGF2alpha-induced airway hyperresponsiveness [50]. Also, in a guinea pig model of ovalbumin-induced chronic allergic airway inflammation, specific airway resistance has been shown to be reduced, in concomitance with the reduction of multiple markers of inflammation and remodeling, by ROCK inhibition [51]. Other studies show the involvement of ROCK in inflammatory pathways linked to airway hyperresponsiveness. Interleukin-17A, a cytokine released by T helper 17 cells, can increase bronchoconstriction induced by methacholine or KCL through RhoA/ROCK2 signaling [52], a finding consistent with ROCK inhibition of MLCP and thus enhanced ASM contractility. In rodent models of asthma, inhibition of ROCK suppresses pulmonary eosinophilia and the release of chemokines and cytokines [53,54,55], extracellular matrix remodeling [49], as well as airway inflammation [51,56,57]. Recently, interleukin-27 has been shown to down-regulate Rho/ROCK and reduce airway inflammation in a mouse model of asthma [58].

ROCK inhibition has been shown to reduce the release of cytokines from peripheral T cells isolated from human subjects [59]. Interestingly, interleukin-13 (IL-13) has been shown to up-regulate RhoA, which is an activator of ROCK [60]. It appears, therefore, that there is a positive feedback loop between cytokines and RhoA/ROCK that is promoted by the inflammatory environment known to exist in asthmatic lungs. Fasudil, a specific ROCK inhibitor, has been shown to reduce the level of cytokines, including IL-13, in ovalbumin-sensitized mice [55]. In the same study, it has also been shown that fasudil-inhibited house-dust-mite-extract-induced MUC5AC expression in airway epithelial cells. Since IL-13 plays a critical role in MUC5AC expression [61,62], it appears that ROCK also plays a role in mucus secretion by regulating IL-13 signaling pathways. Interestingly, the involvement of ROCK seems to vary according to the cell type mediating the inflammatory response; a study in a must cell-dependent model of allergic airway disease shows that genetically reduced expression of ROCK as well as inhibition of ROCK do not affect airway inflammation but reduce airway hyperresponsiveness in ovalbumin-sensitized mice [63]. Taken together, there is strong evidence supporting the hypothesis that over-expression and/or enhanced activity of ROCK is associated with asthma etiology.

## 7. Therapeutic Strategies for Asthma Based on Inhibition of ROCK and Its Downstream Enzymes in ASM

Evidence discussed in the above sections supports a view that an overactive RhoA/Rock signaling is associated with airway inflammation and hyperresponsiveness, thus playing a role in the asthma pathology. These aspects have been covered by many reviews to which we refer the reader [12,13,14,15,16,17,18]. In this section, we will focus on the role of ASM in asthma etiology due to altered RhoA/ROCK signaling that contributes to the muscle’s dysfunction, and more specifically, to the mechanical properties of ASM cytoskeleton. We will also discuss mitigation strategies based on ROCK inhibition.

It has been shown that the ASM from intrapulmonary airways in human asthmatic lungs is hyperreactive and is characterized by an upregulation of many structural proteins [64]. This is consistent with an increase in structural integrity and force transmission capacity in asthmatic ASM. Interestingly, a study by Duan et al. [65] has shown that overexpression of ADAM33 (A disintegrin and metalloproteinase 33) increases the contractility of ASM cells as well as the ASM cell stiffness. The study has also found that these changes can be abolished by H1152, a ROCK inhibitor. These results are consistent with the observation that ASM from human asthmatics is more “robust” in that it is more resistant to strain-induced decrease in contractility [66]. More specifically, it has been observed that oscillatory strain mimicking DIs causes less reduction in contractility in ASM from asthmatics compared with that from non-asthmatics [66]. Because the lack of a bronchodilatory response to DIs is one of the hallmarks of human asthma, restoring the bronchodilatory DI response in asthmatics could be a productive strategy in treating the disease.

Because ROCK signaling is intimately linked to the regulation of cytoskeletal stiffness of ASM [37,38] and ROCK is overexpressed in asthmatics [47], ROCK inhibitors are the natural candidates for weakening the cytoskeleton and making the ASM more amenable to the disruptive effect of oscillatory strains (such as DIs) that reduce the muscle contractility (thus maintaining airway patency). For the same reason, i.e., effect on cell stiffness and overexpression in hypertension, ROCK inhibitors have received much attention in the development of cardiovascular medicine [67]. Besides asthma and hypertension, the therapeutic potential of ROCK inhibitors extends to cancer, erectile dysfunction, glaucoma, insulin resistance, kidney failure, neuronal degeneration, and osteoporosis [68,69]. In fact, at least two ROCK inhibitors (fasudil and ripasudil) are in clinical use in Japan and China [68]. However, because ROCK is an upstream enzyme for many important enzymes in cell signaling, unwanted side effects of ROCK inhibitors are difficult to avoid. One of the systemic side effects of ROCK inhibitor is vasodilation and hypotension [70]. The systemic side effects also lead to local side effects, such as conjunctival hyperemia and subconjunctival hemorrhage [71]. The unwanted systemic side effects of ROCK inhibitors may to some degree be avoided by inhalation of the drug in the treatment of asthma, but the drug effect on the pulmonary circulation will be difficult to avoid. This brings out the importance of using inhibitors of enzymes downstream of ROCK, such as LIMK and PAK (Figure 1), or even further down the signaling pathway [35], such as reduction in N-WASP activation [34]. Inhibition of downstream enzymes from ROCK in the treatment of asthma is a largely unexplored field.

## 8. Combinational Therapy for Asthma Using ROCK Inhibitors and Other Interventions

Experimental evidence has suggested an exciting potential therapy for asthma using a ROCK inhibitor in combination with a bronchodilator. The greatest advantage of using a combinational therapy is that the dosages of the ROCK inhibitor and the bronchodilator can be minimized to reduce side effects. Wang et al. [41] have demonstrated that the combination of low doses of H1152 (a ROCK inhibitor) and salbutamol, each by itself causing insignificant relaxation in ASM, results in a synergistic effect leading to a significant muscle relaxation. It can be envisaged in a future therapy in which asthmatic patients could inhale a mix of nebulized ROCK inhibitor and bronchodilator to relieve asthma exacerbation.

Wang et al. have also demonstrated that the combination of low dose H1152 with small amplitude of length oscillation, again each by itself causing insignificant ASM relaxation, results in a similar synergistic effect that leads to a significant muscle relaxation [41]. The synergy is even more pronounced when the combination of interventions includes H1152, salbutamol, and length oscillation. This three-factor combination augments muscle relaxation to a level greater than 50%, while each intervention by itself causes <5% (statistically insignificant) relaxation [41].

Again, it can be envisaged that a nebulized ROCK inhibitor in combination with a bronchodilator such as salbutamol can be delivered to asthmatic patients through positive oscillatory pressure ventilation. Tidal breathing provides small amplitude oscillatory strain on the ASM. Small-amplitude-high-frequency pressure oscillation superimposed on low-frequency oscillation (mimicking tidal breathing) has been shown to be effective in relaxing ASM [72]. Such a combination of small-amplitude-high-frequency pressure oscillation with low doses of a ROCK inhibitor and a bronchodilator could be an effective therapy for severe asthma. This is a promising strategy within the context of the recently emerged mechano-pharmacological approach in the treatment of asthma.

## 9. Limitations and Future Research Direction

Before ROCK inhibitors can be used as a standard treatment for asthma, more studies on human patients need to be carried out. There have been numerous clinical trials using ROCK inhibitors for treating various diseases (See Table 1). However, no such trials for asthma treatment have been conducted.

In reviewing the trials listed in Table 1, except for treating Raynaud’s disease [79], all trials showed a significant improvement in symptoms when the Rho/ROCK signaling was downregulated by ROCK inhibitors. More importantly, none of the trial studies reported severe adverse effects. However, it should be pointed out that all trials conducted so far have been designed to investigate short term effects (from days to months), it is not clear whether treatment efficacy will persist in the long run and whether severe side effects from ROCK inhibitor treatment will arise in a long-term trial.

It is obvious that an important future direction for asthma research is to carry out clinical trials using ROCK inhibitors to relieve or even prevent asthma exacerbation. Inhalation of nebulized drugs minimizes systemic effects and should reduce side effects. Another way to minimize side effects is to use combinational therapy. As mentioned above, a ROCK inhibitor combined with a bronchodilator (such as salbutamol) dramatically reduces the effective concentrations of both drugs and at the same time achieves more ASM relaxation [41]. Studies also suggest that the synergistic effect of the combinational therapy can be further amplified when a ROCK inhibitor, a bronchodilator, and low-amplitude, high-frequency pressure oscillation are applied simultaneously during mechanical ventilation in the treatment of severe asthma [41,72]. Studies such as these provide theoretical bases for clinical trials. Because ROCK is a relatively upstream enzyme controlling a multitude of cell signaling, targeting enzymes downstream from ROCK (such as LIMK) should be a more sensible choice in future clinical trials.

## 10. Conclusions

ROCK inhibitors are a potential next-generation class of drugs in the treatment of asthma because of their bronchodilatory and anti-inflammatory properties. An additional property of ROCK inhibitors is their ability to reduce ASM cytoskeletal stiffness which is important in restoring the bronchodilatory and bronchoprotective effects of DIs in asthmatic patients. Evidence from in vitro experiments on ASM suggests that a combination of a nebulized ROCK inhibitor with a bronchodilator in the presence of high-frequency positive-pressure oscillation applied to the lung should be more effective than a ROCK inhibitor alone in reversing bronchoconstriction during asthma exacerbation.

## Figures and Tables

**Figure 1 biology-13-00115-f001:**
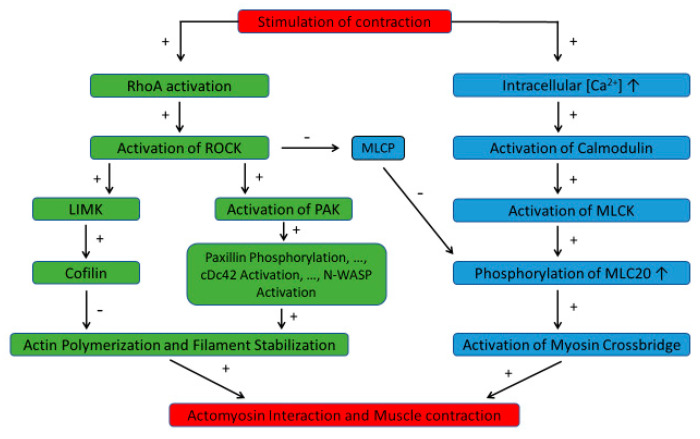
Simplified signaling pathways regulating smooth muscle activation and contraction. The pathway on the right (blue) mainly regulates phosphorylation of MLC20. This is achieved through activation of MLCK by calcium and calmodulin. Once MLC20 is phosphorylated, the myosin crossbridges are able to interact with actin filaments to generate force. The pathway on the left (green) regulates actin polymerization and actin filament stabilization which are important in strengthening the cytoskeleton that facilitates force generation and transmission. The small GTPase RhoA activates ROCK, which in turn activates LIMK leading to inhibition of cofilin and its actin filament severing function. This pathway therefore promotes actin filament formation and stabilization. The other branch of ROCK signaling is through activation of p21-activated kinase (PAK), which leads to phosphorylation of paxillin, activation of the small GTPase cDc42, and N-WASP activation, and eventually actin filament formation and stabilization. More details of the description of the signaling pathways are presented in the main text. +, stimulatory; -, inhibitory; ROCK: Rho-kinase; MLC20: regulatory myosin light chain; MLCK: myosin light chain kinase; MLCP: myosin light chain phosphatase; LIMK: LIM kinase; N-WASP: neuronal Wiskott–Aldrich syndrome protein; PAK: p21-activated kinase.

**Table 1 biology-13-00115-t001:** A list of some clinical trials using ROCK inhibitors for treating various diseases.

Clinical Trials—Relevant Diseases	References
Glaucoma and ocular hypertension	[73,74,75,76,77,78]
Raynaud’s disease	[79]
Pulmonary arterial hypertension	[80,81]
Chronic graft-versus-host disease	[82]
Diabetic macular oedema	[83]
Coronary artery disease and heart failure	[84,85,86]
Crohn’s disease	[87]
Spinal cord injury	[88]
Psoriasis vulgaris	[89]
Acute ischemic stroke	[90]

## Data Availability

Not applicable.

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
