# Peer review of "Rho-Kinase Inhibition of Active Force and Passive Tension in Airway Smooth Muscle: A Strategy for Treating Airway Hyperresponsiveness in Asthma"

_biology, 2024, doi:10.3390/biology13020115_

Round 1

Reviewer 1 Report

Comments and Suggestions for Authors

Summary:

This is a narrative review of the role of Rho-kinase inhibition of tension in airway smooth muscle as a strategy for treating airway hyperresponsiveness in asthma. The review provides a well-organized overview of the signaling models regulating airway smooth muscle, and specifically the role of rho-kinase signaling in asthma. Although it’s not comprehensive, the strengths of this article include the relevance of the topic and the organization of the data presented.

General comments:

The review is well organized as the authors start by introducing the models of signaling pathways regulating airway smooth muscle (ASM) contraction and then move to describe the passive tension in ASM before zooming in on the effect of the over-active ROCK signaling in asthma and the potential use of therapeutics tackling this pathway.

Most of the references are within the last 5 years, with some older ones (most reflecting historical work). There is no excessive number of self-citations.

The figure offers a great visual tool to better understand the two pathways of smooth muscle activation and contraction. More figures/tables can be beneficial in improving readability.  

The conclusion summarizes the review. However, it does not offer an appropriate call for further research by highlighting the knowledge gaps.

There are multiple reviews on the potential of Rho-kinases as a possible target for the treatment of airway hyperresponsiveness. However, this review’s focus on ASM tension is unique.

Specific comments:

38: I prefer defining deep inspiration first, then moving to the differences between asthmatic and non-asthmatic patients.

52-54: please cite an article to back this up.

237: Although not necessary, I think adding a section or table on current/past clinical trials of ROCK inhibitors can improve the last section.

242-245: This sentence is not clear, please re-write for better clarity.

Reviewer 2 Report

Comments and Suggestions for Authors

The manuscript provides a comprehensive exploration of the role of Rho kinase inhibitors (ROCK) in the context of airway hyperresponsiveness and asthma. The authors have presented a thorough review of existing literature, detailing the molecular mechanisms and potential therapeutic implications of ROCK inhibitors in managing asthma.

Major 1:

It is recommended to include a brief section addressing the authors' perspectives on potential scenarios for the application of ROCK inhibitors. Insights into the authors' considerations regarding the standalone use of these inhibitors versus their combination with other medications, such as salbutamol, would significantly enhance the manuscript.

Major 2:

Furthermore, there is a suggestion to incorporate a discussion on the current limitations of ROCK inhibitors and identify areas for future research. Emphasizing the limited number of studies involving patients to date, the authors could express hope that future research endeavors will address this gap by conducting more extensive studies on patients. Acknowledging the challenges and gaps in existing knowledge will pave the way for future investigations and clinical advancements in this field.

Minor:

Lines 78-79: The caption of figure 1 is included in the main text.

Lines 90-93: It is advisable to add a list of abbreviations section for clarity and reader convenience.

Reviewer 3 Report

Comments and Suggestions for Authors

Review on Manuscript Number: biology-2810535

 Rho-kinase inhibition of active force and passive tension in airway smooth muscle: Astrategy for treating airway hyperresponsiveness in asthma

The manuscript presents a description of the action of the class of Rho-kinase inhibitors, which have been identified as a potential class of drugs for the treatment of asthma due to their ability to reduce airway inflammation and active force in airway smooth muscle (ASM). These results may be helpful in determining the suitability of ROCK inhibitors and their downstream enzymes as a class of drugs in the treatment of airway hyperresponsiveness seen in asthma.

The study was conducted to demonstrate the effect of Rho-kinase inhibitors as a class of potential drugs for the treatment of asthma due to their influence on passive tension, which is partially determined by the cytoskeletal stiffness of ASM.

In the first part, called "Summary," the authors describe asthma as a complex disease with multiple causes.

The nature of ASM as the main effector of excessive airway narrowing has been elucidated.

The clear difference in responses between asthmatic and non-asthmatic subjects to deep inhalation (DI) is indicated to have provided us with a unique window of opportunity to examine the mechanisms underlying the pathophysiology of asthma.

The role of Rho--kinases (ROCK) and their influence on passive stiffness of ASM has also been elucidated.

Section 2 contains a detailed description of the canonical models of signaling pathways regulating ASM contraction and cytoskeletal stiffness. Information on an abbreviated version of the signaling pathways regulating smooth muscle activation is also provided.

Section 3 contains detailed information on the two compartments of ASM and their role in asthma relief.

Section 4 elucidates the so-called “passive” tension in ASM and its significance for cytoskeleton stiffness.

Section 5 contains a description of the mechanism of active force maintenance in ASM regulated by ROCK in a mechanically dynamic environment.

Section 6 explains the mechanism of overactive ROCK signaling in asthma, and Section 7 the therapeutic strategies for asthma based on inhibition of ROCK and its downstream enzymes in ASM.

As a result of the review, it is concluded that ROCK inhibitors are a potential next-generation drug class that may be used in the treatment of asthma due to their bronchodilator and anti-inflammatory properties. An additional property of ROCK inhibitors is their ability to reduce ASM cytoskeletal stiffness, which is important to 1.

1.     Overall opinion

The manuscript may be of potential interest to researchers, doctors, and pharmacists, but needs thorough revision before publication to ensure better structure and flow. The text should be revised and organized. A detailed review of the studies cited in the literature is lacking. Clear conclusions are not distinguished. It is necessary to cite more authors to support the said statements.

The English format needs revision regarding word forms and typographical errors in the text.

2.     Comments

The summary is too short. The purpose of the study is not well defined. The abstract does not contain enough information about the practical results of the commented studies and the methods of analysis used in them. There is a lack of information on the methods of application of ROSK inhibitors to regulate ASM functions.

There is a lack of sufficient discussion of the results of the mentioned studies in sections 2, 3, 4, 5, 6 and 7. The degree of credibility of the commented analyzes is also not indicated.

Figure 1 on page 2, line 77 to be colored and the figure title on line 78 to be separated from the text.

In all references from the Referance, the year of publication of the highlighted articles should be bolted.

Do not put the sign ":" in front of the title of the relevant reference in the Referance.

When indicating the name of each author in the references, first indicate the last name and then the first letters of the first name and the dot ".".

Line 462 references are missing information about the specified document, such as issue number, document link, etc.

In the conclusion, there is a lack of sufficient information about the ways of inhibiting ROSK and their combination with other interventions aimed at relaxation of ACM. 

Comments on the Quality of English Language

Review on Manuscript Number: biology-2810535

 Rho-kinase inhibition of active force and passive tension in airway smooth muscle: Astrategy for treating airway hyperresponsiveness in asthma

The English format needs revision regarding word forms and typographical errors in the text.
